# Blockchain for Doping Control Applications in Sports: A Conceptual Approach

Flavio Pinto [1,2], Yogachandran Rahulamathavan [1,*] and James Skinner [2,*]

1 Queen Elizabeth Olympic Park, The Broadcast Centre Here East, Institute for Digital Technologies, Loughborough University London, Lesney Ave, London E20 3BS, UK; f.pinto@lboro.ac.uk
2 Queen Elizabeth Olympic Park, The Broadcast Centre Here East, Institute for Sport Business, Loughborough University London, Lesney Ave, London E20 3BS, UK
* Correspondence: y.rahulamathavan@lboro.ac.uk (Y.R.); j.l.skinner@lboro.ac.uk (J.S.)

**Abstract:** Doping is a well-known problem in competitive sports. Along the years, several cases have come to public, evidencing corrupt practices from within the sports environment. To guarantee fair play and prevent public health issues, anti-doping organizations and sports authorities are expected to cooperate in the fight against doping. To achieve this mission, doping-related data must be produced, stored, accessed, and shared in a secure, tamperproof, and privacy-preserving manner. This paper investigates the processes and tools established by the World Anti-Doping Agency for the global harmonization of doping control activities. From this investigation, it is possible to conclude that there is an inherent trust problem, in part due to a centralized data management paradigm and to the lack of fully digitalized processes. Therefore, this paper presents two main contributions: the concept of a multiorganizational decentralized data governance model and a blockchain-based design for one of the most sensitive data-sharing processes within the anti-doping ecosystem. Throughout this article, it is shown that the adoption of a permissioned blockchain can benefit the whole anti-doping community, creating more reliable processes for handling data, where privacy and security are enhanced.

**Keywords:** blockchain; DLT; doping control; Hyperledger Fabric

## 1. Introduction

With the advent of Bitcoin, the phenomenon of blockchain had its trigger, creating expectations over the potential of cryptocurrencies. More recently, the underlying principles of the technology have proved valuable for several other applications. As a consequence, several initiatives are flourishing in areas as diverse as healthcare, supply chain management, government services, manufacturing, digital rights management, digital identity, energy, and many others [1]. In 2019, Gartner placed blockchain as one of the top ten technological trends, with market researchers suggesting that the technology's value creation may reach the mark of USD 3.1 trillion by 2030 [2].

As discussed throughout this article, organizations participating in the anti-doping (AD) domain have strict needs to collaborate among themselves, in a way to establish trustworthy processes where different instances of data need to be generated, stored, and shared. Given the high stakes involved in competitive sports, humans are sometimes inclined to subvert processes to their own benefit. Additionally, as such processes involve sensitive personal data, all possible protection to the disclosure of identities shall be considered. It is paramount to assure the veracity, immutability, and auditability of produced data, and the adoption of blockchain platforms is becoming a popular solution for such scenarios.

The World Anti-Doping Agency (WADA) was created in 1999 to lead global efforts in the fight against doping. Hundreds of signatories have adhered to the agency's specifications and policies. Among them are the major sport governing bodies and anti-doping organizations (ADOs) at international, regional, and national levels. The agency has created

the World Anti-Doping Code (WADC) and a set of complementary annexes establishing best practices and guidelines. WADA is also responsible for the Anti-Doping Administration & Management System (ADAMS) [3], a centralized web-based database that serves as a clearinghouse for doping control processes and is accessed by members of the participating organizations [4]. ADAMS concentrates different categories of sensitive doping-related data (e.g., whereabouts, doping control reports, therapeutic use exemptions (TUEs) reports, laboratory results, anti-doping rule violations (ADRVs), and intelligence for doping control) [3,5]. The present AD strategy considers a diversification of efforts beyond the traditional detection of prohibited substances from laboratorial exams [5]. Transparent collaboration, reliable, timely data, and secure operational procedures are needed [5,6]. Recent investigations have reinforced previous evidence indicating the widespread use of performance enhancing drugs in sports and found acute processual failures in different organizations, concluding that incumbent siloed, centralized systems for data management create difficulties for a collaborative, fraud-proof, transparent, and secure framework [7]. Therefore, the following research question is posed:

-       How can the AD ecosystem benefit from blockchain technologies for its data governance model and data sharing processes?

The adoption of a blockchain platform can enable decentralized governance for data, and processes can be executed by means of decentralized applications and smart contracts, with all transactions permanently recorded on the decentralized ledger. This work intends to explore how available technologies can be applied to create a more democratic and trustworthy framework for the governance and exchange of data within the AD ecosystem. To answer the research question, investigations were conducted to identify recurring problems within the current AD governance model and processes, as well as a thorough study of blockchain technologies to identify the applicability of existing platforms and features needed to achieve the desired goals. The scope of research executed until present, permits the proposition of an approach for data governance and the design of a sensitive data sharing process, both at the conceptual level. Therefore, the main contributions of this paper are:

-       The definition of a semi-decentralized data governance model for a consortium of organizations, based on a permissioned blockchain, where only authorized entities (organizations and individuals) can participate.
-       A blockchain-based redesign for the request of Therapeutic Use Exemptions (TUEs), a highly sensitive data sharing process within the AD ecosystem, where the benefits of the new approach are clear when compared to the current process.

For the organizations involved in combatting doping in sports, the proposed framework has the potential to disrupt the status quo paradigm for data management. The remainder of the paper is organized as follows:

-       Section 2 explores the relevant literature on blockchain technology and problems pertaining to the current operational model underlying the AD domain.
-       Section 3 discusses a blockchain-based approach for the AD ecosystem, including the adopted research strategy for this work, the suitability analysis describing why blockchain is a good fit for AD, a conceptual decentralized data governance model, and a blockchain-based design for the TUE request process.
-       Section 4 discusses a SWOT analysis for blockchain adoption in the present context.
-       Section 5 concludes the article with indications of future research directions.

## 2. Background

This section is divided into three sub-sections to provide the background of this research. The first sub-section is an overview of blockchain technology, and the second, a description of related work where blockchain technologies are proposed to resolve similar problems to the ones faced by AD organizations. It is assumed that the fundamentals of blockchain technologies are already known by most readers; therefore, these two sub-

sections are brief and intended to simply contextualize the discussion. The third sub-section reproduces the main problems encountered in the current AD operational model, highlighting where new technological approaches can add value.

## 2.1. Blockchain Overview

Since the idea of a peer-to-peer electronic cash system was introduced by the notorious Bitcoin whitepaper in 2008 [8], other distributed ledger technology (DLT) platforms rooted on the same two technological pillars (cryptographic tools and distributed ledger systems) and incorporating additional improvements have been developed to address different business scenarios [1,9–11]. The terms 'blockchain' and 'DLT' are interchangeably used to address platforms gathering a certain set of attributes, namely distributed, decentralized, disintermediated, consensus-based, possessing an immutable ledger to record the history of transactions, and cryptographic mechanisms for the pseudonymization of participants and transactions. Another key functionality incorporated by most platforms is the ability to run Smart Contracts (SCs), which, in essence, are coded business logic that can be enforced autonomously within the network of peer nodes [10,11]. It can be considered that the main value added by DLT platforms is the possibility to enable new trust models, without the need of a single central authority; this arrangement can be disruptive for many business scenarios [10]. Therefore, the traditional centralized model for digital platforms is under scrutiny these days, and proposals for decentralized data governance are regarded as relevant opportunities for transitioning the governance of digital platforms from monopolistic models to more democratic community efforts [12,13].

Another important aspect surrounding DLT platforms refers to the taxonomy of these systems, which is essentially divided into two major categories, permissionless (or public) platforms and permissioned platforms, each having its advantages and disadvantages depending on the envisioned application [1]. In permissionless blockchains such as Bitcoin and Ethereum, any member without distinction can freely join the network, which may be considered a risky situation for certain businesses [9,10]. To address this gap, permissioned blockchains have been designed; these platforms are typically governed by consortiums of organizations needing to achieve common goals without necessarily trusting each other, and willing to benefit from the unique features of DLTs, while restricting the participation to members of a specific business ecosystem [1]. Another important aspect is that, since the participants have interests on a fair trust model that benefits all, a consensus based on financial incentives to validators (as in public blockchains) is not a must. Additionally, since nodes are identified, practical byzantine fault-tolerance (PBFT) algorithms are enough for reaching consensus, which reduces the computational power requirements compared to consensus algorithms of public blockchains [1,14]. Compared to public blockchains, the main advantages are better scalability, lower latency, and higher transaction through-puts [14]. Another benefit is that, if malicious behavior is identified, the misbehaving organization may be deemed responsible, and its participation revoked [1].

## 2.2. Related Work

This research is the first to consider a blockchain framework to address the AD domain; therefore, related work is restricted to other business scenarios where similar problems can be found. As will become clear in next sections, in the AD scenario, two specific issues must be addressed: more effective privacy-preserving data sharing processes, and the traceability and integrity of digital and physical assets. The literature review shows that other business domains are confronted with similar needs. The healthcare sector, for example, is permeated with blockchain case studies [15]. Researchers and analysts have been proposing radical changes in existing business models, moving from vertical siloed or-ganizations to horizontal, integrated, and service-oriented cross-functional structures. The ideal IT infrastructure should integrate innovative products and services, such as ambient assisted living and sensor-based distance health monitoring, with an environment where different service stakeholders (i.e., patients, regulators, healthcare providers, and insurance

companies) can easily collaborate and share records with privacy, in a tamperproof and secure manner [16–21]. Supply chain management (SCM), logistics, provenance assurance, and fraud prevention are other major blockchain applications. In common, they use permissioned blockchains to optimize costs, reduce friction, improve transparency, and assure the origin, authenticity, integrity, and custody of assets throughout their lifecycles [22]. Essentially, these categories of applications are intertwined, as they complement each other and gather similar elements of value. Real-world examples can be found in the agri-food sector [23,24], luxury market [25,26], digital forensics [27], pharmaceuticals [28,29], and international cargo logistics [30], to mention just a few.

*2.3. Analysis of the Current AD Ecosystem*

This section summarizes the main problems encountered in terms of data management within the AD domain. A thorough analysis was conducted so as to obtain a perspective of the social, political, processual, operational, and technological interactions within the current domain. The literature related to AD subject matter includes special reports [5,7], books [31], research articles [32–52], documentaries [53], and official WADA publications (WADC, standards, guidelines, press releases, and statistics) [54–72].

It has been observed that the effectiveness of AD efforts is repetitively questioned by the scientific community. This affirmation is ratified by the low official statistics of violations over the years compared to the estimated prevalence of doping [4,51]. Despite the declared commitment of organizations and stakeholders with a doping-free ecosystem, several scandals reported in literature indicate that dishonest participants may not be deterred from utilizing or supporting doping practices and finding gaps to cheat AD policies [7]. Hence, the sports ecosystem is permeated with a tacit conflict of interests [6,7,31]. Competitive athletes, trainers, and sport organizations, in their will for victory, control organizations, trying to guarantee the fair play and eliminate the problem of doping [7,31].

This work does not intend to discuss anti-doping in its entirety or suggest that a new technological approach would be a panacea. Nevertheless, DLTs can certainly play a role in guaranteeing more trustworthy processes and less centralization. By identifying the vulnerabilities of current systems and processes, new approaches can be proposed to difficult cheating attempts. The main challenges in terms of data management are summarized hereafter:

**Siloed/Centralized Decision Making and Databases:** From a political standpoint, the way WADA deals with anti-doping activities is still considered monopolistic, as the agency keeps most of the decision making with itself or delegated to a restricted number of international sport federations [52]. Each organization manages its own AD database. With WADA being the global authority for doping control, ADAMS should be fed with all doping-related data produced by the various participating organizations, but in practice, it does not happen effectively [5,7]. From 2015 to 2017, two committees of the UK parliament conducted an inquiry that relied on oral and written evidence, academic research, investigative journalism, and whistleblowers. The findings are described in a report published in 2018, where acute failures in several different organizations have been found: failure in sharing appropriate medical records with anti-doping organizations; failure in keeping proper internal records of the medical substances given to athletes; and failure in outlawing the use of potentially dangerous drugs in certain sports [7]. Reported cases indicate that AD data are often kept within their silos and not shared appropriately. One of the several examples presented in the cited report describes that an important international federation (IF) did not share a few thousands of suspicious test results with the relevant ADOs, and the data were kept isolated in the IF's data silo and used by corrupt members for extorting athletes. As a result, potential ADRVs could not be pursued by ADOs [7]. Concurrently, a point of concern occurs when data are transferred to ADAMS. As ADAMS gathers data belonging to the different stakeholders, but is controlled by a single organization, WADA, this asymmetry is viewed as too much power concentrated on a single organization, a potential conflict of interests, and a barrier for a consistent trust model [5,50]. A central

database is also a single point of failure; cyber-attacks have already occurred, with athletes' sensitive data being leaked into public domain [49,67,71].

**Non-Digitalized Processes:** Some important processes are still relying on paperwork or semi-digitalized processes. Relevant examples are the collection and the chain of custody (CoC) of doping control samples, the TUE request, and the Athletes' Whereabouts [4,56,57,63]. Non-digitalized processes are more easily circumvented or frauded, with no evident signs being left behind. One of the most notorious scandals occurred in the 2014 Sochi Winter Olympic Games. A state-sponsored, audacious, and large-scale fraud scheme for swapping compromised urine samples and tampering with positive results was established, and only revealed later thanks to one of the participants who decided to blow the whistle on the scheme [7,68,69]. To make these processes safer, the adoption of digital technologies, such as biometry for identity assurance, electronic forms, IoT sensors, and other forms of digitalization, can play an important role.

**Insufficient Anonymization and Privacy:** To perform their tasks and following a selective security criterion, organizational staff have privileged access to records [5]. Although participants of the AD program assume the responsibility to comply with the International Standard for the Protection of Privacy and Personal Information (ISPPPI) [4,58], trust is totally on the individuals, making it difficult to trace any eventual leakage of personal data. Another example is the samples' CoC, as each bottle is anonymized, meaning the athletes' identities are not informed to the laboratories. However, a bottle's apparent identification number is informed to the athlete; then, if corrupt members within the organizations are inclined to trade in favor of athletes willing to clean their samples, bottle swapping schemes can be established, as already mentioned in the Sochi Winter Olympics case [68]. The same is valid for the TUE request; once the submission is undertaken, the identity of the requestor will become known at different checkpoints in the process by internal staff within the involved organizations [7,68,69]. The adoption of digital technologies for anonymization of physical or digital assets can be helpful to avoid privacy leaking issues.

Figure 1 exemplifies two of the current processes within the AD ecosystem, doping control and TUE request, highlighting some of their vulnerabilities.

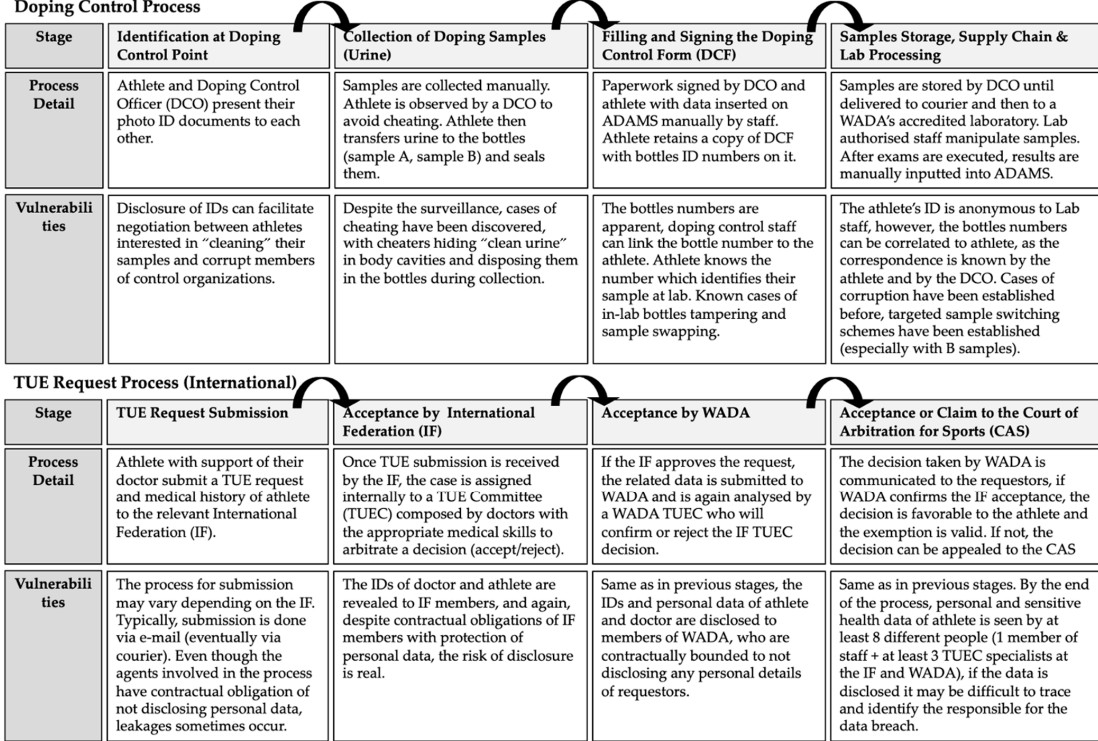

**Figure 1.** Examples of the current doping control and TUE request processes and their vulnerabilities. References: [7,31,55–58].

## 3. Blockchain-Based Approach for the AD Domain

This section covers the defined research strategy and main contributions of the paper. The research strategy is described in Section 3.1, and it clarifies the present research with its outcomes and contributions, as well as the future research scope and its expected outcomes and contributions, including methodological approaches. Section 3.2 analyses the suitability of blockchain for AD, posing relevant questions and respective answers confirming it. Section 3.3 presents the design of a blockchain-based approach, divided into two sub-sections, the first (Section 3.3.1) covering governance aspects of a consortium effort involving key organizations within the AD community, and the second (Section 3.3.2) presenting the design of the TUE request process, highlighting clear improvements in terms of data assurance and privacy preservation. These are regarded as the main contributions of this paper.

### 3.1. Research Strategy

The research strategy can be visualized in the form of a workflow presented in Figure 2. This research can be categorized as a scenario analysis, per the definitions discussed in [73]. The research strategy until the present has been strongly focused in conducting literature review, which followed a logical sequence. First, varied aspects of doping were investigated by reading a curated list of publications. As a second step, investigations focused on the official WADA documents. As a result, based on the solid comprehension of the AD domain, the problem statement was established. The third stage consisted in acquiring a technical comprehension of blockchain technologies, in terms of applications, suitability of the technology to the examined scenario, and design aspects. The fourth step can be defined as a conceptualization stage, where the outcomes correspond to the contributions highlighted in this paper. The future scope of research includes a technical validation, simulating the conceptual design [14,74], and the quantification of the impact of this research to the AD/sports ecosystem by the application of a Delphi study [73,75]. The future research scope is expected to create two additional contributions to be submitted for future publication.

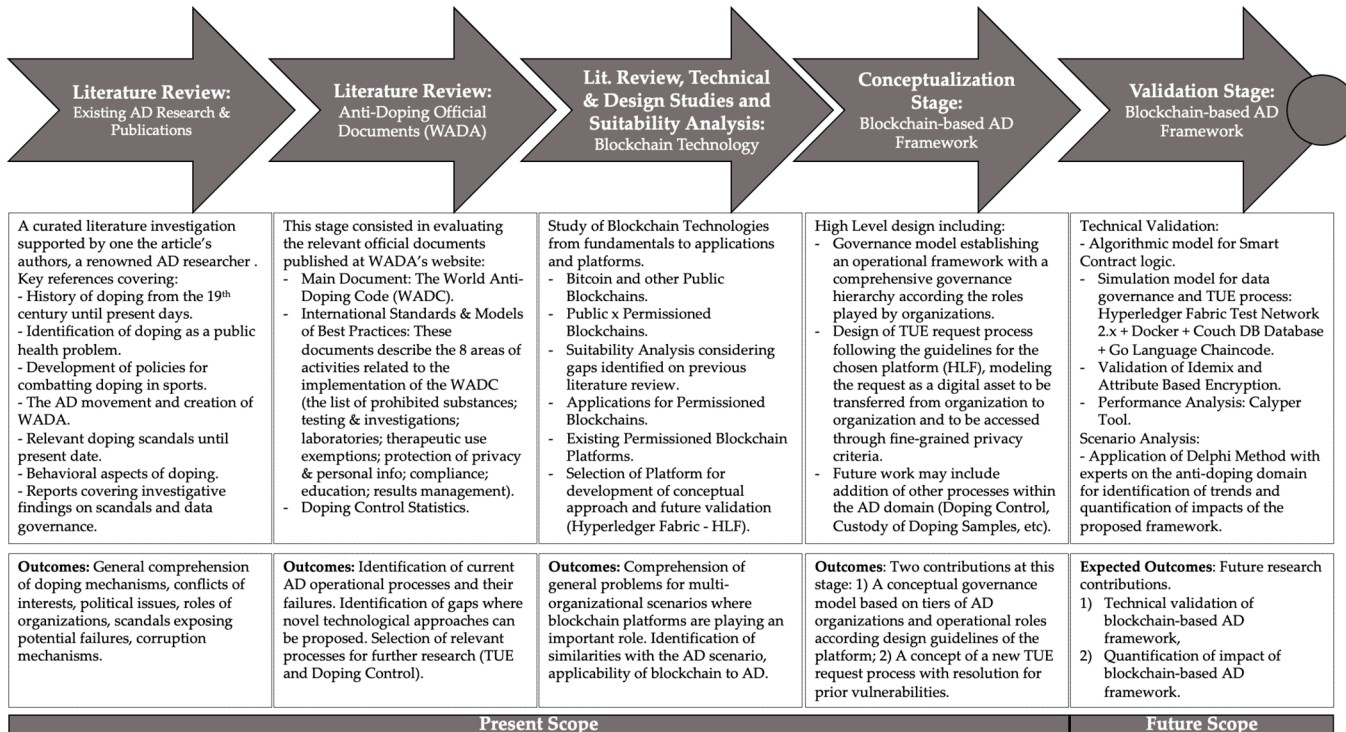

**Figure 2.** Research strategy workflow.

### 3.2. Suitability Analysis

Organizations are currently investigating the potential of blockchain for different business scenarios, but its adoption should not be seen as a must do process due to the hype created by the technology [76]. A blockchain architecture can add complexities and increased costs due to its distributed nature and strong reliance on cryptographic and consensus schemes; therefore, merely adopting it without a prior investigation of the value that can be added can lead to frustrations [1,76]. The recommended approach is to first understand the technology, its strengths and weaknesses, and then identify existing or new ecosystems that can clearly benefit from this decentralized paradigm [1]. For business scenarios with just one or few participating organizations, non-distributed participants, no transactional data, no problems with a centralized trust model, no conflicts of interests, no problems with reconciliation of data, no need for full provenance of digital assets [1], blockchain may not add enough value to justify a change [76]. The analysis, then, needs to be performed for each specific situation, and to support this initial assessment, different decision-making models are available in the literature. Although these models differ on their formulation, the core factors for the suitability of blockchain are common in all models. This suitability analysis is formulated based on the available literature [1,76,77], with relevant questions applicable to the AD scenario posed and answered in Table 1. From this analysis, it is possible to conclude that the business problem investigated in the AD ecosystem gathers enough elements to benefit from a permissioned consortium blockchain.

**Table 1.** Blockchain suitability analysis.

| Criteria | Justification |
| --- | --- |
| 1. Does the application need multiple parties accessing the network and writing data to a decentralized database? | Answer 1: Yes, in the current business model, individuals from multiple organizations access read and write data to a centralized database (ADAMS), which is governed by a single organization (WADA) [5,50]. A decentralized shared database can improve the trust model and reduce frictions within the AD domain [1]. |
| 2. Can a single Trusted Authority in control of the data store lead to a potential conflict of interest? | Answer 2: Yes, currently, ADAMS is controlled by a single authority (WADA). This model creates an asymmetry of information and a conflicting trust model for organizations, as confirmed by the literature [5,50]. Therefore, a decentralized approach can disrupt this model, considered monopolistic and unfair by several participants [52]. |
| 3. Are data immutability and integrity of transactions desired? | Answer 3: Yes, records include doping control tests, therapeutic exemptions, athletes' whereabouts data, or other sensitive data pertaining to doping control activities [5]. This kind of data should not be modified, or if there is a need for update/modification, it should be part of an amending process that can be audited and traced by the AD authorities [5]. The integrity of the produced and stored data is key for creating a fully traceable history of transactions. |
| 4. Is there a need to keep the content of transactions private? | Answer 4: Yes, the content of AD transactions must be kept confidential and be accessed only by authorized parties and according to specific access criteria [5,58]. In this context, permissioned blockchains are preferable, as only authorized parties can participate in the blockchain and fine-grained privacy control of transactions can be implemented [14]. |
| 5. Is consensus determined inter firm? Are the writers known and trusted? Is it important to control functionality? | Answer 5: Yes, inter-firm consensus with known and trusted writers and controlled functionalities, pointing to permissioned blockchains [1]. Inter firm consensus promotes democratization, which is a desired feature for the AD domain [52]. Writing rights are given only to trusted members of the consortium. Functionalities, policies, and membership might be changed over time; therefore, the control of functionalities is essential [78]. |
| 6. Is the required application feasible without high throughput for transactions? | Answer 6: Yes, high throughput is not critical for the AD domain. An order of magnitude for the needed throughput can be taken from this example: during the 2016 Rio Olympics, the busiest day had a total of 350 doping samples collected [79]. If all data related to these controls were inputted at the same time, it would still be supported by a platform such as the Hyperledger Fabric (up to 20,000 transactions/second) [80,81]. |

*3.3. Design of a Decentralized Anti-Doping Ecosystem*

Considering the analysis towards the adoption of a permissioned blockchain, platform-specific articles, design guides, and books were checked [14,74,78,82]. Hyperledger Fabric (HLF) was the chosen platform for this work. HLF is an open-source project governed by the Linux Foundation, possessing a modular architecture that enables confidentiality, flexibility, and scalability and allows components such as the consensus and membership services to function as plug-and-play accessories [74]. HLF modularity can be described horizontally by identity management, ledger management, transaction management, and smart contracts components, and vertically by membership management, consensus services, chaincode services (chaincode is the proprietary name given to SC implementation in HLF), and security and cryptographic services components [83]. All nodes in the network are identifiable, enabling a controlled environment. The concept of consensus follows an execute–order–validate process, which eliminates uncertainties in the formation of blocks. Another important characteristic of HLF is its ledger technology, which is decomposed in two parts, a blockchain log and a state database. The blockchain log stores the history of transactions; in other words, the state transitions of an asset, which is a write-only process, and each block recorded is a collection of transactions. The state database is a representation of the current state of an asset, which allows some performance improvement for the system; its absence would require computer programs to traverse the whole transaction log to calculate the asset state. In HLF, the assets stored in the state database are defined as key-value pairs, which can be created, erased, or updated; therefore, the state database can change frequently [9,14,74]. The privacy protection, anonymity, and security of the ledger can be handled by the platform's cryptographic services and digital certificate management system [14,74]. In any blockchain system, different aspects of privacy shall be considered: Transactional Data Privacy; State Data Privacy; SC Privacy; and User Privacy (which refers to the anonymity and unlinkability of users). These aspects are not properly addressed by permissionless blockchains, but in HLF three first aspects are implemented by default, while the aspect of user privacy requires additional attention [74,83]. The main aspects of this conceptual approach are discussed in what follows.

3.3.1. Semi-Decentralized Platform Governance

The platform governance is mainly defined in terms of the governance model, the organizations' roles, and the operational policies [78,80]. In the AD case, a consortium-led model is assumed, in which the leading organizations (e.g., WADA, major ADOs, and major Sports Authorities) in the domain come together and define the platform's policies and the participants' roles. A recommended procedure is to move the blockchain network ownership and operation under a single legal entity, which can be a joint venture (JV), maintained and controlled by the co-founding members. The JV equity holders can appoint a board of directors, representing the interests of each member organization. Then, the board itself can appoint an executive management team in charge of the JV operations. Board meetings can occur regularly to deliberate governance decisions, which then can be implemented at system level. Two distinct roles are defined for the JV [78]: Network Service Provider (NSP) and Business Service Provider (BSP). The NSP role includes managing and enforcing the agreed platform policies. The BSP role includes the development and deployment of the platform's business logic (applications, smart contracts, and integration). With this model, one entity is accountable for harmonizing the interests of its various shareholders and community members in general. This setup avoids legal, regulatory, and intellectual property issues [78]. At first, the platform would gather a few key members of the domain, with non-profit goals, being primarily focused on better harmonizing the AD standards. At later stages, other organizations can be allowed to join the ecosystem. Reviews of the JV ownership and economic and management models could occur as a natural progress of the relationships within the AD domain, and the technology is flexible to reflect such changes [14,78]. The governing members decide which organizations are allowed to participate in the platform's decisions, enabling a democratic governance. The

policies contain, among other things, the lists of organizations with access to a specific resource or how many organizations must digitally vote for updating a resource, such as a SC or a channel [14,78]. Apart from the role played by the governing JV, the other organizational roles needed to achieve the desired results are [78]:

**Ordering Organizations (OOs):** Organizations hosting ordering nodes can vote for certain policies, such as the consensus model and consortium membership. These organizations form the so-called Ordering Service, which distributes blocks to the network.

**Network Service Consumers (NSCs):** Organizations that host Peers (with smart contracts and copies of the ledger) and Certificate Authorities (CAs). These organizations have the capability to vote for business logic (e.g., SCs and channel membership) and endorse transactions.

**Business Service Consumers (BSCs):** Organizations that host client-side applications and manage blockchain identities of users.

**End Users (user level):** These are the individuals who will connect to the platform with different access privileges set by the organizations and respecting the governance policies.

The idea behind the different organizational roles is to create a decentralized hierarchy, where the more roles an organization assumes, the more committed it is with harmonization of the WADC and AD standards, and the more active is its participation in the governance of the blockchain. Figure 3 is divided in two parts: Figure 3a depicts the key types of organizations with governance status composing the AD consortium and Figure 3b details the proposed governance hierarchy with four levels, their roles, and the types of users managed by each hierarchical level. The highest level is the Governing JV, and as previously explained, this is the organization that takes the responsibility for the blockchain's operation. The Governing JV is the only organization with NSP and BSP roles and it can also play the three other roles down in the hierarchy (OO, NSC, and BSC). Then, the second level is composed of the so-called 'Governing Organizations'; they include the OO, NSC, and BSC roles. These are the major organizations directly involved in combating doping, such as WADA, the major ADOs at national and regional levels, the major International Federations, and the International Olympic Committee (IOC). The next level down the hierarchy is for 'Trusted Organizations', and organizations in this level will include the NSC and BSC roles, being able to validate transactions, participate in consensus, and host copies of the ledger, but are not authorized to participate in governance decisions and in the ordering of blocks. Lastly, the lowest level is for 'Client Organizations', which only act as clients in the ecosystem; they do not have a permanent role in the ecosystem, therefore participating occasionally in some processes and not being allowed to host SCs or copies of the ledger.

### 3.3.2. Network Architecture

The described governance structure implies a general blockchain architecture as exemplified in Figure 4. HLF supports the creation of different channels, and each channel has its own immutable ledger to record the history of transactions [74]. Only organizations authorized to participate in a channel obtain access to the ledger [14]. The channel feature is very convenient for the AD scenario, as the different processes can be logically isolated from each other. In this paper, a specific process within the AD domain was addressed, the request for Therapeutic Use Exemptions (TUEs) [56], which will be explained in the next sub-section. The following logic sequence was used when operationalizing a distributed application in HLF.

*Channel Creation:*

When a channel is created for a specific application, it should include the relevant organizations that have interest in benefiting from the blockchain process. An important aspect to observe is the management feature of the system. The network initiator (in this case, Organization 1-JV Organization) has authority to set up the network configuration (NC), which is defined at the initial ordering service (O1).

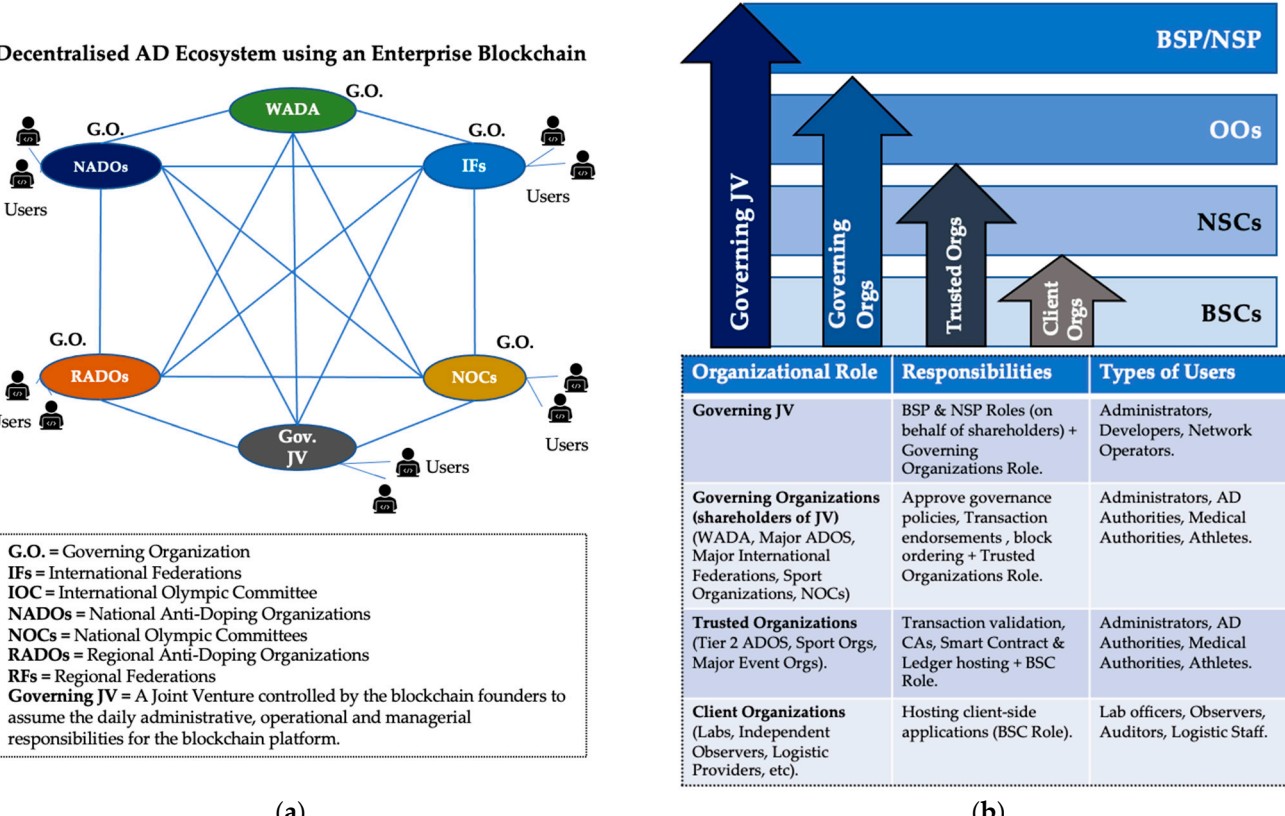

**Figure 3.** (**a**) A decentralized multiorganizational model for the AD ecosystem. (**b**) The governance hierarchy proposed for a decentralized AD ecosystem. Reference: [78].

*Channel Configuration:*

As a second step, channel configurations (CCs) are defined. NC and CCs contain the policies agreed by the consortium members for the network as a whole, and for each channel individually. In the discussed example, it can be noticed that Organization 1 is the only one that has administrative rights at the network level (NC). Organization 1 created two channels, Channel 1 (which hosts the TUE application) and Channel 2 (other application or process can be created in a separate channel with its own rules). In Channel 1, organizations 2 to 5 were added, and organizations 2, 3, and 4 have 'Governing Organization' status (administrative privileges on the channel per the policies defined in CC1), while organization 5 is on a level below, acting as a 'Trusted Organization' (therefore not being able to vote for governance policies on the channel). The policies can be changed anytime upon agreement of the governing entities. In general, the participating organization with a trusted status administers at least one peer (P) node, where a copy of the ledger is hosted as is one certificate authority (CA) or a membership service provider (MSP) for issuing X.509 or Identity Mixer identities to all blockchain elements and users, and for mapping them to their corresponding member organization.

*SCs Installation and Instantiation:*

Smart contracts are installed on the peers and instantiated on the channels [74]. A business process initiated at a client application invokes a smart contract to access the ledger. Considering the HLF 'execute–order–validate' approach, transactions are proposed, executed, and endorsed before being packaged into a block. The endorsement policy can be defined to have a minimum number of endorsements; at least the organizations directly involved in the request should endorse the transactions. After a block is delivered to all peers, transactions within the block are validated by checking their consistencies with endorsement policies [14]. The block is then committed to the peers' copies of the ledger

and a block event is generated. Therefore, all organizations participating in a channel will have at least one copy of the ledger (organizations can have more than one peer for extra redundancy of data). A Database modelled on a ledger has guaranteed integrity (data are immutable and tamperproof) [14].

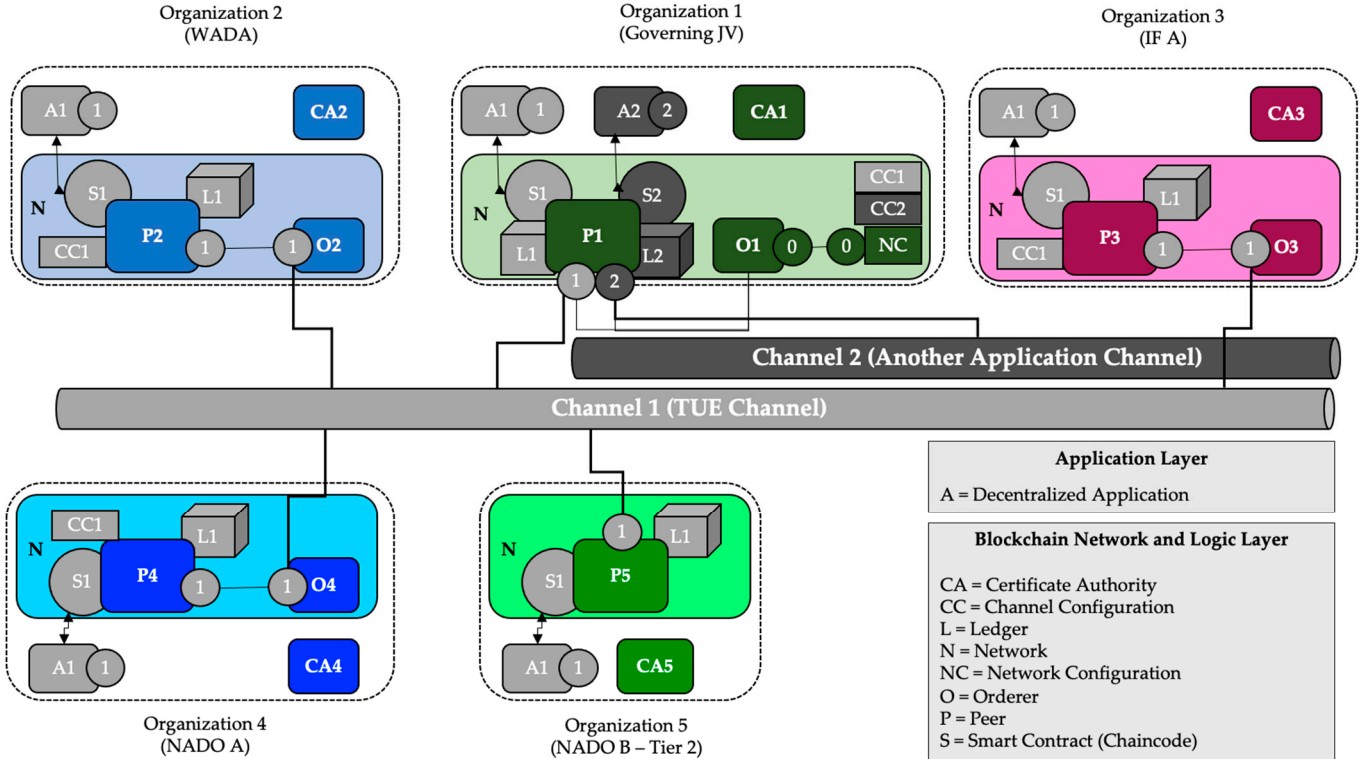

**Figure 4.** General network architecture for consortium application in HLF, with an example with different channels for different applications. References: [14,74].

### 3.3.3. Chaincode and Asset Lifecycle Design

In Section 2.3, different vulnerable data sharing processes within the AD domain were exemplified. In this paper, the focus was placed on one of these processes, the request for a Therapeutic Use Exemption (TUE). A TUE may be granted to an international athlete, for example, after being submitted to the scrutiny of the medical committees at the International Federation (IF TUEC) and WADA (WADA TUEC) [56]. If granted by both organizations, the athlete is allowed to use medication that otherwise would be forbidden due to possible doping effects. If not granted, the athlete can appeal to the Court of Arbitration for Sports (CAS) [56]. The data submitted and the outcomes within this process are kept as confidential as possible [56,58]. The reason for choosing this specific process over others is that, as suggested by the AD literature, athletes, their doctors, and trainers show enormous concern in sharing medical data with others through the existing process [5,49,50]. The current process for submitting and obtaining a TUE is not fully harmonized, meaning that different organizations may adopt different procedures, and the data are exchanged via paperwork-based forms and e-mail-based communications, resulting in the insufficient anonymization of athletes' data. The data can often leak or be disclosed inappropriately due to such vulnerabilities within the process [5,49,50].

With a fully digitalized approach and blockchain, the same process can be redesigned to enhance data integrity, assurance, security, and confidentiality. The TUE request process can be mapped as a SC (Chaincode) logic in HLF [84]. The Chaincode must be designed and coded with the purpose to validate each stage of the TUE request. The Chaincode is installed and instantiated in the peer nodes participating and endorsing the process. From this perspective, the TUE becomes a digital asset in HLF that has a clear lifecycle

(a sequence of stages). At each stage, the participating organizations contribute with data and the Chaincode is invoked via the decentralized application to modify the asset's attributes accordingly. The following logical sequence is proposed:

1. Submission Stage: The athlete's doctor submits the TUE request to the applicable International Federation (IF). The asset is transferred by the doctor to the IF, and this transference is a transaction recorded on the blockchain.

2. Receipt Stage by IF: The TUE request is received by the IF at the administrative level. The TUE request details can be analyzed by the IF administrative members with the correct credentials (the doctor's and the athlete's identities should not be disclosed). This allows the IF to determine the IF TUEC medical competences needed to properly evaluate the TUE request. Once the competences are defined, the IF determines the IF TUEC composition. The asset is transferred to the IF TUEC to be analyzed by their members; the transaction is recorded on the blockchain.

3. Receipt Stage by IF TUEC: The IF TUEC receives the asset, performs its analysis, grants, or rejects the request. The analysis and decision are recorded, and the asset is transferred to the next organization, in this case WADA. The transference transaction is recorded on the blockchain.

4. Receipt Stage by WADA: The asset is received by WADA at the administrative level. WADA administrative members with the correct credentials can view the asset's content, the IF TUEC analysis and decision, and determine their own WADA TUEC. The asset is transferred to the WADA TUEC; the transaction is recorded on the blockchain.

5. Receipt Stage by WADA TUEC: The WADA TUEC analyses the IF TUEC response, performs, and records their own analysis and decision (uphold or revert the IF TUEC decision). Once the WADA TUEC decision is taken, the asset is transferred again, with the transaction being recorded on the blockchain. Depending on the WADA TUEC decision, two different outcomes can occur.

6. Accepted Stage by Doctor (Alternative 1): If the IF TUEC decision is upheld, the asset is finally transferred back to its original owner, the athlete's doctor. The asset is transferred back to the doctor, so as to evaluate the decision and take further actions if needed. In the case that the TUE is granted both by the IF and WADA TUECs, the athlete and the doctor will just accept it and may need to share results with other organizations. If the TUE is rejected by both TUECs, the doctor and the athlete may decide to appeal the decision with the Court of Arbitration for Sports (CAS).

7. Rejected Stage by IF TUEC (Alternative 2): It the IF TUEC decision is to grant the TUE, but the WADA TUEC rejects it, the asset is transferred back to the IF, and they may decide to appeal against the decision with the Court of Arbitration for Sports (CAS).

From the above-described sequence, it becomes clear that, for each transaction transferring the asset from one stakeholder to the next, the asset transition is recorded on the blockchain, creating a traceable and permanent history for future audits. The asset information of each individual stage is directly related to the Chaincode logic; therefore, it provides the functionality to store data of each stage, query existing data, and complete the lifecycle history for any given TUE request. Throughout the process, the personal details of the athlete and doctor are supposed to remain anonymous; sensitive data preservation schemes will be explained in the next section. The TUE asset format for each stage is shown as a class diagram in Figure 5.

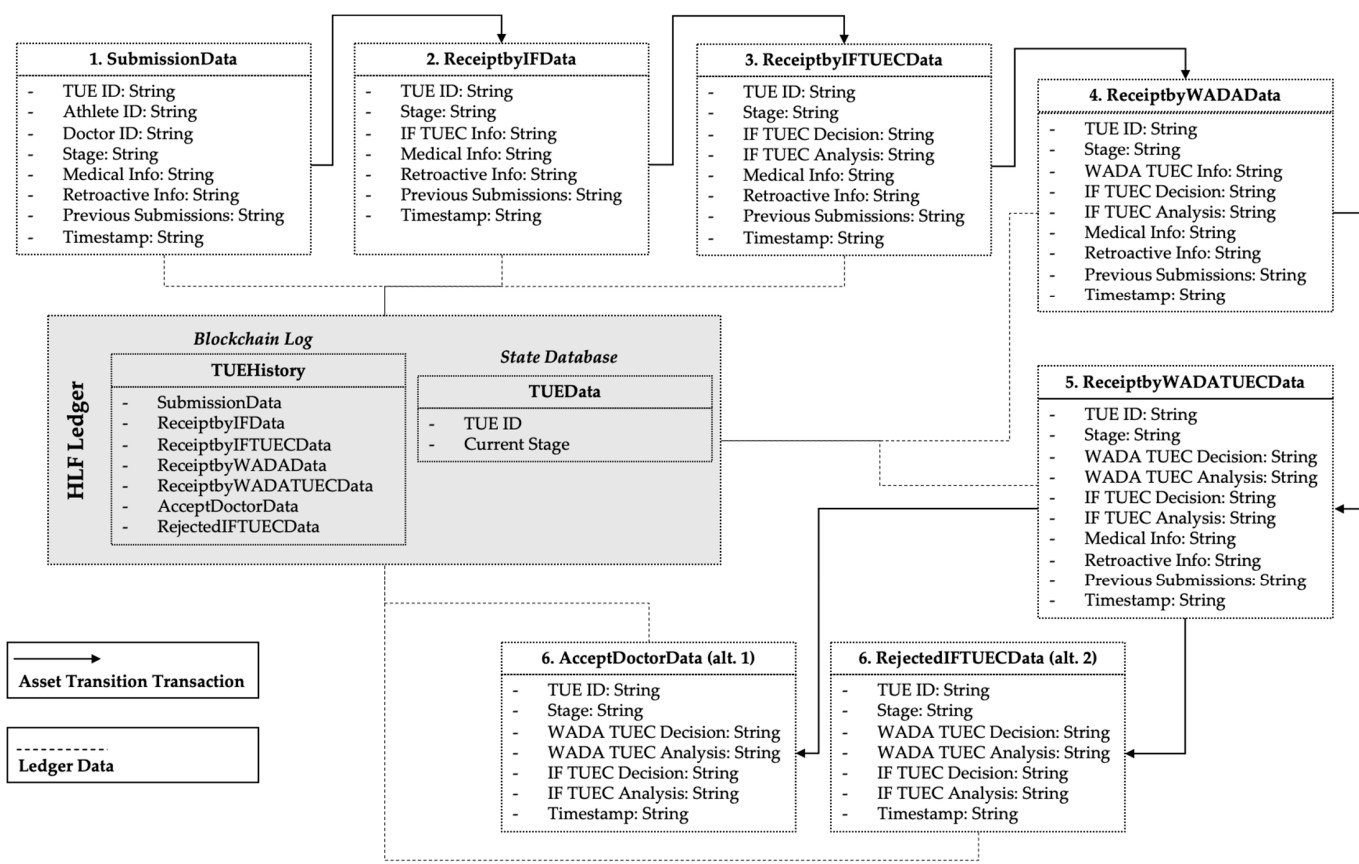

**Figure 5.** TUE request–asset lifecycle. References: [83,84].

### 3.3.4. Membership, Access Control, and Confidentiality

As discussed in the opening paragraph of Section 3.3, HLF is a permissioned platform where only pre-authorized stakeholders can participate. This functionality is assured by the chosen access mechanisms and channel settings; hence, it is the combination of a Membership Service Provider (MSP) and an Access Control List (ACL) that enables privacy and security in HLF [83]. The participating organizations connect with each other through the HLF network, and each organization has at least one peer node that must deal with membership services to their users. The standard implementation is not sufficient to guarantee the anonymization of users' data as envisioned in this proposed application; therefore, more sophisticated pluggable schemes can be proposed. Attribute-based encryption (ABE) and Identity Mixer (Idemix) are advanced techniques used to address privacy and confidentiality of users and users' data. The ABE scheme provides confidentiality and access control with a single encryption. In this paper, the distributed ABE framework proposed in [85] was used. The ABE scheme involves the following parties: the MSP, the User Application (data input), the Attribute Authorities (AAs), and the Distributed Ledger. AAs are implemented within the MSPs of trusted organizations only (in accordance with the governance hierarchy proposed in Section 3.3.1), which are responsible for verifying and issuing credentials for users based on their attributes. The distributed ABE scheme was chosen so as to avoid a single point of failure. In this case, different AAs issue credentials for users, allowing different sets of attributes to be monitored by different AAs. Figure 6 shows the ABE functionalities and a possible multiorganizational distribution considering the various organizations in the AD ecosystem and the proposed governance hierarchy.

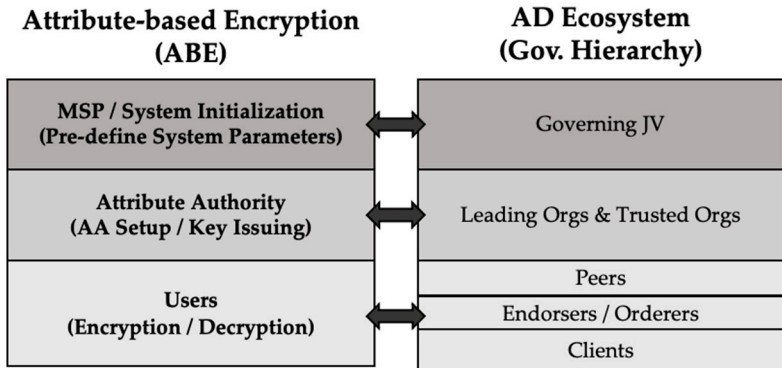

**Figure 6.** Mapping of ABE functionalities/AD organizations. Reference: [85].

The ABE operational scheme is described by five protocols [85]:

1. Setup: By this protocol, a predefined parameter of security is taken as input. The system parameters are the outputs generated by the Governing JV MSP and shared only with organizations with AA responsibility.
2. AA Setup: Using the system parameters obtained from setup, each AA organization can generate public and private keys for the attribute it maintains.
3. Key Issuing: Via an anonymous key issuing scheme, user (i.e., peers, orderers, and clients) and organizations interact so as to determine a set of attributes belonging to the users. Then, the AA organization generates decryption credentials for those attributes and sends them to the user.
4. Encryption: The encryption algorithm is used by the blockchain users at the application level. The user takes a set of attributes maintained by AAs and the different segments of the 'TUE request' data as inputs. Then, it outputs different ciphertexts of each data that will be appended in the transaction.
5. Decryption: The decryption algorithm is also used by blockchain users. After taking the decryption credentials received from AAs and the ciphertext from the blockchain transaction as inputs, the decryption will be successful if, and only if, the user attributes satisfy the access structure of the ciphertext.

Figure 7 shows the input format of a TUE request, the structure of the initial blockchain transaction and examples of ABE policies for data decryption. At the user application side, the request is submitted via an electronic form that has three distinct segments of data: the athlete's personal data, the doctor's personal data, and the TUE-related details (i.e., sport, medical conditions, and prohibited substance(s)). Each segment of data in the form is encrypted with a specific ABE policy, and the appropriate set of attributes for each segment is defined in accordance with the TUE request criteria. The encrypted data are then appended to the blockchain transaction invoked via SC. Since the transaction is published in the ledger, it is visible to participants in the channel; however, as the sensitive personal data of involved parties and the details of the TUE request are encrypted with ABE, only the users possessing the right set of attributes can decrypt them. Members of TUECs, for example, will receive keys that only allow them to view the TUE-related data, but not the athlete's nor the doctor's personal details. Similarly, the personal information of both athlete and doctor could only be decrypted by specific authorities with auditing attributes, for example. Therefore, ABE encryption policies allow fine-grained access to sensitive data [85], guaranteeing privacy and confidentiality during the analysis of a TUE request, which is a considerable gain compared to the current semi-digitalized and centralized process and its vulnerabilities (as explained in Figure 1). Figure 7 only shows the application of ABE for the initial submission of the TUE request; however, it is important to highlight that analogous ABE criteria will be applicable to the following stages of the asset lifecycle.

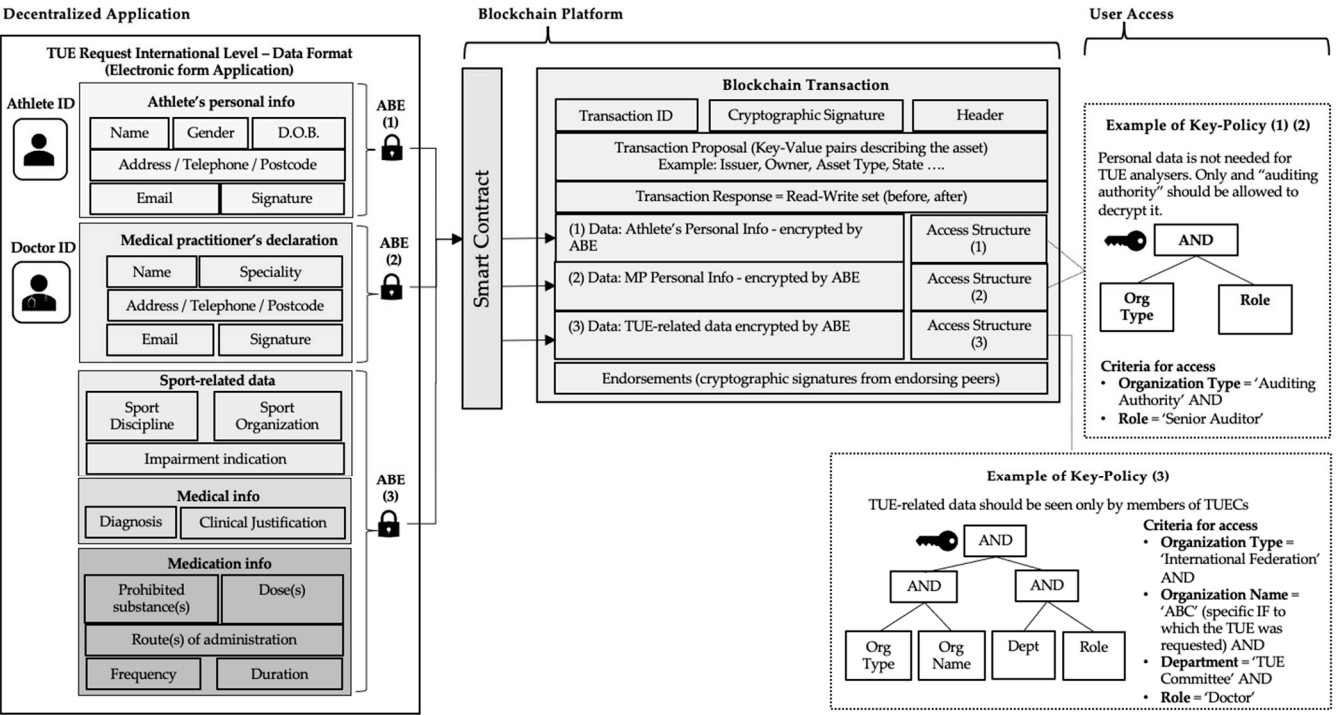

**Figure 7.** Data format (TUE request)/transaction structure/examples of ABE policies.

To complete the proposed scheme, the MSP implementation counts with Idemix to play an important role. HLF's basic MSP implementation complies with a standard Public Key Infrastructure (PKI), which does not support users' anonymity and unlinkability. With Idemix, user anonymity and unlinkability are guaranteed and the transaction can be invoked without disclosing the transactor's identity. A single identity can transact several times without disclosing that the transactions were sent by the same identity [83,86]. Issuer MSPs in the proposed scheme certify a set of user's attributes that are issued as digital certificates (also known as "a credential") [86]. The user later generates a zero-knowledge proof (ZKP) of possession of the credential, and only discloses the attributes previously chosen to be revealed. The proof, because it is zero-knowledge, reveals no additional information to the verifier, issuer, or to any other participant [86]. When a blockchain user needs to transact, a certificate must be obtained from a Certificate Authority (CA), so an MSP interacts with the CA to issue certificates, including an enrolment certificate. When a user in the blockchain network invokes a SC transaction, that transaction needs to be previously signed by the enrolment certificate [83]. With Idemix, the user can generate per-transaction certificates; in other words, certificates that are used just for one transaction. The same principle is valid for encryption keys, one-time-use keys are created, and the user can prove all keys were derived from the same enrolment certificate [83]. This guarantees to the user the desirable ability to generate multiple keys from the original enrolment certificate, but at the same time being able to prove to an auditor in the HLF network that all transactions were performed by a single user, while other entities in the network would not be aware that all transactions were performed by a specific user [83]. Considering the TUE request example, this means that the athlete's doctor can initiate the process by invoking the TUE request transaction, while keeping their identity anonymous to the other stakeholders via ZKP and encrypting all sensitive data with ABE. It is important to highlight that Idemix is not yet mature in the current HLF release [86], and ABE is not a standard feature; therefore, a complete functionality combining a decentralized ABE scheme and Idemix represents a considerable development effort.

3.3.5. Security and Privacy Aspects

HLF is a platform designed to enable secure collaboration between organizations [87]. In the AD scenario, all participating organizations have social responsibilities towards the community [32]; therefore, it is unlikely that participants would engage in malicious behavior and put their reputations in risk. Nevertheless, threats can potentially occur, which makes important a work for identification and definition of mitigation strategies [87]. The following aspects are listed:

Consensus and Endorsements of Transactions: The most common consensus threat for public platforms, 51% of attacks, is not a considerable issue for permissioned platforms [1,9,10]. In the context of HLF, the current consensus process only utilizes Crash Fault Tolerant (CFT) algorithms, where zero tolerance against malicious actors is established [87]. Nevertheless, monitoring key indicators, such as the latency of transactions and leadership elections of transactions' endorsers [14,74], can be a strategy for the early detection of malicious behavior [87]. If only authorized participants access the ledger, the adversary model considers the attacker is a member of the channel [87,88]. When a user proposes a transaction, preferred endorsers can be chosen; therefore, the identities of the chosen endorsers are disclosed to participants of the channel, which is because in the current HLF version, Idemix cannot yet be used for endorsement of a SC transaction or for approval of SC definitions [86]. Assuming the existence of an adversary, a selective denial of service (DoS) attack can be executed by the adversary, in an attempt to modify the content of transactions, which can be a treat for any distributed system [87,89]. A motivator for the adversary would be to defeat the endorsement policy requirement embedded in the SC [87,89]. As an example, consider an adversary attacks a certain number of endorsement responses to the user. If a user proposes a transaction and sends it to N endorsers and the policy within the SC requires at least (N-4) endorsements, if the attacker dumps or modifies five responses, only (N-5) endorsed responses are sent back to the user, causing the transaction proposal to fail. Repeated DoS attacks can also degrade the overall network efficiency [87]. A risk mitigation strategy for DoS attacks is the proactive collection of performance metrics [87]. The wormhole attack is another threat to transaction's endorsements [89]. Within a channel, a group of identities can be separated from the rest and every member has access to the ledger. If a single member acts maliciously, colluding with an adversary outside the channel, the attack can be executed. In this case, a virtual private network (VPN) with the outer adversary leaks internal information from the consortium without any knowledge of honest members [89]. Two strategies for mitigation are proposed. The first strategy focuses on anonymizing the endorsers to protect against DoS attacks. This is achievable by two different techniques. The first considers the randomization of endorsers by use of Verifiable Random Functions (VRF) implemented in the SCs. The second proposes the pseudonymization of endorsers to protect against wormhole attacks. To achieve this, the anonymization of transactions within a channel is needed. The sender is anonymized in a first step and the receiver in a second step [89]. Overall, test results suggest that immunity against DoS and wormhole attacks were achieved, with a tradeoff on the network efficiency [89].

Smart Contracts: SCs must be designed with precise logic that encapsulate the defined business process and mitigate common bugs and errors in handling nondeterminism and concurrency [87]. Additionally, best practices of software lifecycle management must be embedded in the governance policies for the platform, as proposed in Section 3.3.1, where SCs can only be deployed after approval by all SC executors. The current version of HLF embeds decentralized SC lifecycle policies to facilitate the governance process [74]. IT tools to analyze SC logic can also help in the detection of anomalies before SCs are deployed [87].

MSPs: MSPs handle the access control to the network, and if captured by a malicious insider, could be used to deny service and execute sybil attacks, posing a serious threat [87]. This risk can be mitigated by adoption of best practices in key management [14,85,87]. In the ABE encryption scenario described in Section 3.3.4, the identity management process is dependent on attributes managed by different organizations; the adoption of this distributed approach also helps in minimizing the risks, as no single MSP manages all

attributes for the keys [85,87]. Finally, a strategy for managing the logs of MSPs can help to detect malicious behavior and avoid MSPs to become compromised [87].

## 4. SWOT Analysis

SWOT analysis is a popular strategic planning and management tool applied in industry and academia to help the understanding of Strengths, Weaknesses, Opportunities, and Threats related to business project planning and competitive landscapes [90]. Recognizable by most business readers, it provides a simple structured approach to evaluate the strategic position of a project or organization, by identifying the internal assets (the inherent strengths and weaknesses) and comparing these to the external factors, defined as the opportunities and threats in the environment [90,91]. When applied to information technology (IT) projects, SWOT may help in answering important strategic questions [91], such as: What do the project's stakeholders expect to achieve? How can the project be distinguished from the status quo? How will the project improve delivered services? How can it be assured that the IT strategy complements strategic organizational goals?

In the context of this article, the SWOT formulation is applied to evaluate the perspectives of the specific blockchain framework proposed for the AD domain, as discussed hereafter.

Strengths:

Data Integrity, Immutability, Traceability, and Auditability: These strengths are listed together as they are the direct result and the most obvious strengths of the basic schemes implemented on any blockchain platform. Attempts to alter or tamper with validated data in the blocks are detected, and due to the distributed and hashed nature of sequential recordkeeping, are nearly impossible to achieve [1,9]. The ledger is replicated in several nodes and can be accessed by the different members of the network; therefore, auditability is achieved, but with certain rules for access (not fully transparent as in permissionless blockchains) [1,9,10].

Data Quality: Considering that produced data are the result of carefully coded and consensually validated business processes, the data are considered high-quality. Blockchain data are consistent, complete, accurate, timely, and widely available [88].

Semi-Decentralized Governance: While the fully decentralized model of permissionless blockchains can be considered (with regard to data governance) a disadvantage, the semi-decentralized model of permissioned consortium blockchains is perceived as a strength for the scenarios where they are applied [13]. The concept of a shared governance model, where members of a consortium can establish powerful modes of democratic collaboration without losing control over the platform's rules and policies, is what makes them attractive to many industries [1,9,10]. It is also suggested that the semi-decentralized mode of data governance enhances the overall performance of the platform, compared to the fully centralized or fully decentralized options [13].

Privacy Protection: As discussed in previous sections, privacy protection can be implemented by different and combined schemes of cryptography. A benefit to users is that permissioned blockchains enable the possibility for more customized, individualized and fined-grained control over access to personal data [74,83].

Efficiency and Integrity of Business Processes: The combination of peer-to-peer architecture, SCs, and consensus enables automated trust in networks where the participants may not fully trust each other [88]. It is possible to implement processes for the enforcement of contractual agreements and reconciliation of data, eliminating intermediaries and gaining efficiency compared to traditional centralized IT infrastructures [1,9,88,92]. Because SCs are executed in different peer nodes with the same outputs, the integrity of processes is guaranteed [88].

Different Ledgers to Different Applications: One key feature of HLF and a strength for the AD scenario is the possibility to create multiple channels [14,74], to address different business processes within the domain. For each business process, a specific set of participants and specific sets of rules and policies need to be established, and the logical isolation

between them facilitates coordination and contributes to security and privacy within the ecosystem [58].

Weaknesses

Potential Non-Compliance with Data Laws: The proposed AD blockchain framework addresses a global community of stakeholders with a semi-decentralized approach for data governance. The removal of a single centralized authority, the transnational aspect of the network of peers, and the append-only permanent data storage are features that may challenge the traditional legal frameworks [93]. Data localization laws are sometimes established at sovereign level, imposing limitations for data replication beyond national borders [92]. Requirements imposed by the General Data Protection Rules (GDPR) and the "right to be forgotten", for example, can be ensured by throwing decryption keys after the data retention time is elapsed, or by assuming off-chain storage schemes for personal data with on-chain pointers, which would allow the actual data to be deleted in accordance with time-elapsed rules. However, there's no consensus that such remedies fully meet GDPR requirements [92,94]. The regulatory framework for the adoption of blockchain is not yet fully defined or harmonized at global scale [93].

Architecture Complexities and Lack of Standards: Compared with traditional databases, the architecture of blockchains inserts certain complexities for developers and operators [88,94]. For example, logical data layer models in blockchain environments require a developer to consider two constructs, assets and SCs, with differences from platform to platform [94]. In conventional databases, this layer is a well-defined and standardized area of application programming [88,94]. In the physical data layer, blockchains are divided in three tiers: transactions, blocks, and ledger, but there is no standard, and the internal organization and data structures of these tiers are platform-dependent [94]. In the data access layer, reading data is not straightforward and the immutability feature challenges the development of alternatives for allowing data deletion [88,94]. More efficient data access to the application layer is a critical request for blockchain-based systems, with on-going efforts in this area aiming to support faster and more sophisticated queries [94].

Performance and Scalability: Balancing on-chain and off-chain storage is an important consideration for most scenarios. For big data applications, on-chain storage may lead to increased cost and performance loss due to scaling the replicated data stores. Therefore, hybrid approaches for separating data are commonly adopted [94,95]. Typically, the application's hashes are kept on-chain and actual data are stored off-chain. This configuration provides better performance and scalability while still achieving immutability (data hashes allow integrity verification of off-chain data) [96]. On the other hand, it introduces challenges for aggregating and integrating data from different sources [94]. From a data administration perspective, permissioned blockchains allow on-premises or cloud installations. The platform's choice, its parameters, hardware configuration, workload, and mode of hosting have a significant impact on performance. Therefore, a recommendation is to benchmark and fine tune the platform before production use [94].

Long-Term Preservation of Data: The long-term preservation of blockchain data is still an open question [92]. A general assumption when adopting blockchains is that the replicated data store will automatically guarantee the long-term preservation, security, and availability of records, but this perception might not always be correct [92]. With regard to the AD scenario, it is assumed that organizations are interested in keeping their ledgers, which is a measure to guarantee their own interests with respect to eventual disputes. Nevertheless, establishing a minimum commitment in terms of time and quantity of nodes to keep the integral version of the ledger is an essential measure that must not be overlooked.

Integration with Legacy Systems: There is no doubt that the eventual adoption of a blockchain framework must consider the integration with legacy systems, which is generally reported as a complex and laborious process [88].

Opportunities

Technology Momentum: The adoption of blockchain for different applications is now considered a reality, with worldwide spending increasing at a Compound Annual Growth Rate (CAGR) of 62.35% from 2017 until 2021 and last year's spending of USD 6.6 billion [97]. A 2021 survey also indicates that secure information exchange is currently the main use case with 45% of respondents indicating business interest on it, and digital currencies coming second, with 44% of respondents [97].

Innovate Business Processes: The proposed blockchain AD framework has not been seen in previous academic studies, and its impact is yet to be measured as part of future research scope (see Section 3.1). Nevertheless, this article thoroughly describes how changes in the governance model and data exchange processes can redefine trust in the system. Related work in other business segments with similar needs, such as healthcare, is abundant with positive transformations and consolidation has been recently reported [15].

Threats

Governance Disputes: One of the big challenges for scaling distributed ledgers is related to controversies around governance issues. With so many variants and differences in each scenario, best practices are not clear yet [96]. Decentralization, disintermediation and democratization are key appeals for blockchain adoption; however, a fully decentralized governance may be problematic in many cases [13]. In consortium cases with a semi-decentralized governance, as discussed in this paper, different organizations must abide by a mixture of industry standards and statutory requirements related to how data are managed and secured [96]. These specific requirements may push DLT/blockchains to be tailored in unique manners. Additional challenges are posed when extrapolating to the international level with different socio-cultural values to be harmonized [96].

Costs of Implementation and Adoption: The costs involved in developing and implementing the AD blockchain framework tend to be high; systems are not fully mature yet and intense research continues to create more standardized approaches [88]. Additionally, the participating organizations have different levels of financial resources and interests in joining the system [6,7,31]. As described at the beginning of this article, despite the institutional commitment to adhere to the WADC [54], and the benefits of the proposed blockchain approach, there is no guarantee that all members of all organizations will automatically approve the change.

The SWOT analysis is summarized in Table 2.

**Table 2.** SWOT analysis of proposed AD framework.

| Strengths | Weaknesses |
|---|---|
| Data Integrity, Immutability, Traceability, Auditability<br>Data Quality<br>Semi-decentralized Governance<br>Privacy Protection<br>Efficiency and Integrity of Business Processes<br>Different Ledgers to Different Applications | Potential Non-Compliance with Data Laws<br>Architecture Complexities and Lack of Standards<br>Performance and Scalability<br>Long-Term Preservation of Data<br>Integration with Legacy Systems |
| **Opportunities** | **Threats** |
| Technology Momentum<br>Innovate Business Processes | Governance Disputes<br>Costs of Implementation and Adoption |

An effective SWOT analysis recommendation is to, whenever possible, identify how strengths can match with opportunities, and how weaknesses can be converted in strengths, and threats in opportunities [90]. The first weakness 'Potential Non-Compliance with Data Laws' can be addressed by representants of the AD domain via an early involvement in regulatory discussions, aiming to create new regulations or change existing ones, to facilitate technology adoption within the domain [93]. The weaknesses 'Architecture Complexities and Lack of Standards', 'Performance and Scalability', and 'Integration with Legacy Systems' are currently being addressed by scientific research and practical business

implementations in several business segments, in a trial-and-error process that will produce knowledge and new insights. Given the current status of global adoption [97], blockchain technology seems to have passed the hype stage and chances of success are high. The strengths of the proposed AD approach are a definitive match with the opportunities at the present moment; blockchain technology is in clear expansion and applications for 'secure information exchange' are perhaps the most popular of them all [97]. Finally, the identified threats can also be considered part of the maturation process of this novel technology. 'Governance Disputes' must be addressed by clear definitions at design stage and adoption of an operational model that will minimize the chances of conflict; however, as addressed in Section 3.3.1, the most important factor is the flexibility of the technology to allow changes in the governance model whenever needed, which is the case in HLF [78]. As for the 'Costs for Implementation and Adoption', the tacit conflict of interests indicated in AD literature [6,7,31] and discussed in Section 2.3 may be a significant threat; however, if a blockchain framework is adopted by leading sports and AD organizations, the others will most likely follow it [32,52], sooner or later.

## 5. Conclusions and Future Work

This article examined the idea of integrating a permissioned blockchain platform to disrupt current data management and information sharing processes of AD in professional sports. As described in the research strategy section of this article, the present research scope had outcomes grounded on two literature review stages focusing on the AD scenario (Doping in Sports and WADA Operating Model/Harmonization of AD Practices), followed by a literature review integrated with the technical research of blockchain technology, applicability of blockchain to the investigated AD scenario, and platform-specific technical and design studies stage. The research conducted in these three preliminary stages culminated in the present research contributions highlighted in the conceptualization stage. The thorough analysis of the AD domain revealed the existence of chronic trust problems, with a history of corrupt acts, circumvention of doping control processes, and tacit conflicts of interests both in terms of the goals expected by doping control organizations and with regard to data governance. Evidence to back up these affirmations is abundant in the existing body of research related to doping in sports. The suitability analysis confirms that the AD domain fits into a permissioned blockchain approach, where a semi-decentralized governance operating on top of a consortium-based cooperation scheme creates more streamlined, efficient, and trustworthy data management processes, with additional positive impacts on democratization of data and less friction between participating organizations. To provide a conceptualistic vision of a decentralized ecosystem for the AD domain, HLF was the chosen platform; therefore, a model of data governance was proposed based on its operational characteristics and in accordance with a natural existing hierarchy of organizations interacting in the AD ecosystem. By analyzing WADA official documents, it was also possible to identify the exact processes where data needs to be shared amongst different organizations and individuals, and where vulnerabilities due to non-digitalized or partially digitalized processes leave gaps for cheaters or result in weak privacy of sensitive personal data and consequently loss of trust in the system. One of these processes, the request for a TUE, was scrutinized and conceptually redesigned as a sequential SC business logic that creates a digital asset with a clear lifecycle. Each stage of the asset is described by an immutable transaction recorded on the blockchain. In this paper, the current gaps and vulnerabilities of this specific process were clearly exposed, and the proposed redesign based on a blockchain approach demonstrates exactly how each of these gaps and vulnerabilities are addressed, resulting in concrete benefits for involved organizations and individuals. Specific cryptographic features for creating a fine-grained access control of personal data recorded on the blockchain and anonymization of identities (ABE and Idemix) were proposed as key elements of the proposed design. Security aspects of the system and strategies to mitigate them were also discussed in the paper. Finally, the SWOT analysis for adoption of blockchain in the AD domain was discussed, indicating a

positive perspective, which however is not free of challenges and that need to be addressed appropriately. Overall, this article concretely demonstrated how decentralization and the inherent features of blockchain combined with advanced cryptographic features can enable the profound transformation of data governance within the AD domain.

This research has already a future scope clearly defined, which will be submitted for future publications as the research progresses. The future research scope consists of validating the concept here described through technical simulations and measuring the impact of blockchain via a Delphi methodology, allowing the final combined scope of research to create a robust scenario analysis of blockchain applied to AD in sports. Although not included in the formal research scope defined in the research strategy section, other interesting future research opportunities for blockchain in AD can be explored. It is important to highlight here that the TUE request addressed in this article is just one of the processes within the AD domain. Building on top of the proposed architecture and governance solution, other processes could also benefit from this framework. The most obvious example is the CoC of doping control samples. This process currently shows clear gaps in the way the samples are handled and in how the anonymization of sensitive personal data is implemented; therefore, a fully digitalized supply chain approach on the blockchain with a clear business logic mapped in the SCs, where sensitive personal data are properly anonymized and the physical assets (the doping sample bottles) are digitally tracked all the way through the CoC, would provide additional value to the AD domain.

**Author Contributions:** Conceptualization, F.P., Y.R. and J.S.; methodology, F.P., Y.R. and J.S.; formal analysis, F.P., Y.R. and J.S.; investigation, F.P.; resources, F.P., Y.R. and J.S.; writing—original draft preparation, F.P., Y.R. and J.S.; writing—review and editing, F.P.; supervision, Y.R. and J.S.; project administration, Y.R. and J.S. All authors have read and agreed to the published version of the manuscript.

**Funding:** This research received no external funding.

**Informed Consent Statement:** Not applicable.

**Data Availability Statement:** No applicable.

**Conflicts of Interest:** The authors declare no conflict of interest.

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
