# Peer review of "Blockchain for Doping Control Applications in Sports: A Conceptual Approach"

_futureinternet, doi:10.3390/fi14070210_

Round 1

Reviewer 1 Report

Line 265 - Why? Could you provide more explanations about the businesses for which doesn't fit?

Table 1 - A summarization should be explained and references should be provided for each of the item from the table.

Figure 1 - It is created by the authors? If not, please cited properly.

Figure 2 - idem

Figure 3 - idem

Line 512 - ABE Encryption description of the algorithms should be improved signficantly. It is not clear what are the inputs parameters and which is the output.

Security analysis should be improved accordingly. No advantages and security games are described. What is the probability?

Author Response

Reviewer 1 

Dear reviewer,  

Thanks for your valuable comments and suggestions for improving the manuscript. Your comments have been addressed accordingly and responded point by point as follows.  We are maintaining a proactive reviewing process, to identify any further improvements that can be addressed, even before additional feedback is received about this revised manuscript. We believe considerable improvement was already achieved with this reviewed version, but more can still be done.

Open Review 

English language and style 

( ) Extensive editing of English language and style required  
(x) Moderate English changes required  
( ) English language and style are fine/minor spell check required  
( ) I don't feel qualified to judge about the English language and style 

Yes 

Yes 

Can be improved 

Not applicable 

Does the introduction provide sufficient background and include all relevant references? 

( ) 

(x) 

( ) 

( ) 

Are all the cited references relevant to the research? 

( ) 

(x) 

( ) 

( ) 

Is the research design appropriate? 

(x) 

( ) 

( ) 

( ) 

Are the methods adequately described? 

( ) 

(x) 

( ) 

( ) 

Are the results clearly presented? 

( ) 

(x) 

( ) 

( ) 

Are the conclusions supported by the results? 

( ) 

(x) 

( ) 

( ) 

Comments and Suggestions for Authors 

Line 265 - Why? Could you provide more explanations about the businesses for which doesn't fit? 

Answer: Thanks for this suggestion. We have included additional explanations in the beginning of old sub-section 3.1., now sub-section 3.2 (a new sub-section to highlight the research strategy was added in 3.1.). The new manuscript highlights why blockchain is not necessarily fit for all business scenarios. A blockchain adoption inserts complexities (peer-to-peer decentralized architecture, consensus algorithms, heavy reliance on encryption algorithms) and certain challenges (scalability, lower transaction throughput), therefore its adoption may not make sense for all business scenarios. Essentially, blockchain will make sense whenever there are several different parties with underlying conflicts of interests, and when trust can’t be relied to a single authority. Also, transactions’ immutability and auditability for all community members are desired features. Here is one extract of text added to the reviewed section 3.2. “A blockchain architecture can add complexities and increased costs due to its distributed nature and strong reliance on cryptographic and consensus schemes, therefore, merely adopting it without a prior investigation of the value that can be added, can lead to frustrations [1][72]. The recommended approach is to first understand the technology, its strengths and weaknesses, and then identify existing or new ecosystems that can clearly benefit from the blockchain paradigm [1]. For business scenarios with just one or few participating organizations, non-distributed participants, no transactional data, no problems with a centralized trust model, no conflicts of interests, no problems with reconciliation of data, no need for full provenance of digital assets [1]; blockchain may not add enough value to justify a change [72]”.

Table 1 - A summarization should be explained and references should be provided for each of the item from the table. 

Answer: In the reviewed manuscript, supporting citations have been included for all items in the table, and explanations extended to make the decision model and the justification for adoption of blockchain clearer. Example of text added to table 1: “Answer 1.: Yes, in current’s business model, individuals from multiple organizations access, read and write data to a centralized database (ADAMS), which is governed by a single organization (WADA) [5][49]. A decentralized shared database can improve the trust model and reduce frictions within the AD domain [1]”.

Figure 1 - It is created by the authors? If not, please cited properly. 

Figure 2 – idem 

Figure 3 – idem 

Figure 4 – idem 

Figure 5 – idem 

Answer: All figures have been elaborated by authors to describe specificities of the Anti-doping scenario. Some of the figures, however, were inspired by existing literature cited throughout the paper. Therefore, whenever applicable, the relevant citations have been added to the figures appropriately.  

Line 512 - ABE Encryption description of the algorithms should be improved significantly. It is not clear what are the inputs parameters and which is the output. 

Answer: Thank you for raising this question. In the revised manuscript, the ABE scheme description has been reviewed and improved accordingly, so to include more details about its objective and functionalities. We, however, understand the underlying mathematical theory of ABE, is complex and beyond the scope of present paper, where our intention is to describe a concept of blockchain as a supporting technology to improve data sharing processes within the AD domain. Moreover, we have directed the advanced readers to the authors previous work [82] which contains more technical details about the amalgamation of ABE with blockchain.

Security analysis should be improved accordingly. No advantages and security games are described. What is the probability? 

Answer: Thank you. This section has been renamed as ‘Security and Privacy Aspects’ and reviewed to include additional considerations, enriching the discussion of privacy and security aspects of the Hyperledger Fabric Platform. This section was designed to cover a simpler discussion of the proposed scheme and adopted blockchain platform. We have not yet been able to develop an analysis to the level of security games theory and probabilistic analysis, but your comments are greatly appreciated and as mentioned above, we are maintaining a continuous and proactive improvement approach to the manuscript, hoping that further improvements can be incorporated to this specific section, for a next review of the manuscript, in case interest is maintained in its publication.

Reviewer 2 Report

The article is clearly written. The key objectives can be highlighted.

Section 2.1, the blockchain technology is well known. Instead, it can be replaced by the similar research, as well as the traditional AD process.

Section 2.2, the meaning of domain analysis is not clear. The references numbering can be replaced by [35-55] and [57-75]. The traditional AD process can be illustrated with some figures.

The methodology and design for the suitability analysis should be elaborated.

English proofreading is required.

Author Response

Reviewer 2 

Dear Reviewer,

Thanks for your valuable comments and suggestions for improving the manuscript. Your comments have been addressed accordingly and responded point by point as follows.  We are maintaining a proactive reviewing process, to identify any further improvements that can be addressed, even before additional feedback is received about this revised manuscript. We believe considerable improvement was already achieved with this reviewed version, but more can still be done.

Open Review 

English language and style 

( ) Extensive editing of English language and style required  
(x) Moderate English changes required  
( ) English language and style are fine/minor spell check required  
( ) I don't feel qualified to judge about the English language and style  

Yes 

Can be improved 

Not applicable 

Does the introduction provide sufficient background and include all relevant references? 

( ) 

(x) 

( ) 

( ) 

Are all the cited references relevant to the research? 

( ) 

(x) 

( ) 

( ) 

Is the research design appropriate? 

( ) 

(x) 

( ) 

( ) 

Are the methods adequately described? 

( ) 

(x) 

( ) 

( ) 

Are the results clearly presented? 

( ) 

(x) 

( ) 

( ) 

Are the conclusions supported by the results? 

( ) 

(x) 

( ) 

( ) 

Comments and Suggestions for Authors 

The article is clearly written. The key objectives can be highlighted. 

Answer: In the reviewed manuscript, the key objectives have been highlighted in the new Introduction section. Here is an extract: “Scope of research executed until present, allows the proposition of an approach for data governance and a design for one of the sensitive data sharing processes, both at conceptual level. Therefore, the main contributions of this paper are:

- The definition of a semi-decentralized data governance model for a consortium of organizations, based on a permissioned blockchain, where only authorized entities (organizations and individuals) can participate.

- A blockchain-based redesign for the request of Therapeutic Use Exemptions (TUEs), a highly sensitive data sharing process within the AD ecosystem, where the benefits of the new approach are clear when compared to the current process.

For the organizations involved in combatting doping in sports, the proposed framework has the potential to disrupt the current paradigm for data management”.

Section 2.1, the blockchain technology is well known. Instead, it can be replaced by the similar research, as well as the traditional AD process. 

Answer: This comment is greatly appreciated. Because blockchain technology is known by most readers, the section has been compressed and simplified to the minimum necessary for contextualization purposes. With regards to the AD process, we have enriched the discussion and added a new figure which highlights the current processes and their vulnerabilities.

Section 2.2, the meaning of domain analysis is not clear. The references numbering can be replaced by [35-55] and [57-75]. The traditional AD process can be illustrated with some figures. 

Answer: The ‘domain analysis’ term has been removed from the sub-section title, which is now called ‘Analysis of Current AD Ecosystem’. A figure has been added to exemplify two current AD processes and the existing vulnerabilities that shall be addressed by the blockchain framework. The references have been merged as suggested. Thanks for this comment. 

The methodology and design for the suitability analysis should be elaborated.  

Answer: We have added a new sub-section 3.1, which presents the research strategy. The suitability analysis was expanded, with justifications re-elaborated and references added. 

English proofreading is required. 

Answer: Thank you. We have now revised the manuscript to improve the readability and we strongly believe that the latest version is better than the previous submission. 

Round 2

Reviewer 1 Report

The authors managed to comply with the comments and provided a significant improvement version of the paper.

Author Response

Dear Reviewer,

Thanks for your comments. This latest revision includes some changes applied to Section 3.3.2. with the purpose to improve readability. Section 4 was fully re-written to include a SWOT analysis, a suggestion offered by the Academic Editor. In addition to these two main changes, only some minor language corrections were implemented throughout the article. 

Kind Regards!

Reviewer 2 Report

comments are addressed.

Author Response

(The authors gave the same response as above.)
